# Infra-slow brain dynamics as a marker for cognitive function and decline

**Shagun Ajmera**
Centre for Neuroscience
Indian Institute of Science
Bangalore
ajmerashagun@gmail.com

**Shreya Rajagopal**
Centre for Neuroscience
Indian Institute of Science
Bangalore
shreyakr96@gmail.com

**Razi Ur Rehman**
Computer Science and Automation
Indian Institute of Science
Bangalore
razirmp@gmail.com

**Devarajan Sridharan**∗
Centre for Neuroscience &
Computer Science and Automation
Indian Institute of Science
Bangalore
sridhar@iisc.ac.in

## Abstract

Functional magnetic resonance imaging (fMRI) enables measuring human brain activity, *in vivo*. Yet, the fMRI hemodynamic response unfolds over very slow timescales (<0.1-1 Hz), orders of magnitude slower than millisecond timescales of neural spiking. It is unclear, therefore, if slow dynamics as measured with fMRI are relevant for cognitive function. We investigated this question with a novel application of Gaussian Process Factor Analysis (GPFA) and machine learning to fMRI data. We analyzed slowly sampled (1.4 Hz) fMRI data from 1000 healthy human participants (Human Connectome Project database), and applied GPFA to reduce dimensionality and extract smooth latent dynamics. GPFA dimensions with slow (<1 Hz) characteristic timescales identified, with high accuracy (>95%), the specific task that each subject was performing inside the fMRI scanner. Moreover, functional connectivity between slow GPFA latents accurately predicted inter-individual differences in behavioral scores across a range of cognitive tasks. Finally, infra-slow (<0.1 Hz) latent dynamics predicted CDR (Clinical Dementia Rating) scores of individual patients, and identified patients with mild cognitive impairment (MCI) who would progress to develop Alzheimer's dementia (AD). Slow and infra-slow brain dynamics may be relevant for understanding the neural basis of cognitive function, in health and disease.

## 1 Introduction

Functional magnetic resonance imaging (fMRI) is among the few techniques that enable non-invasive measurement of activity in the living human brain (*in vivo*) [1]. Although state-of-the-art fMRI scans can measure activity at high (sub-millimeter) spatial resolution, the fMRI blood-oxygenation level dependent (BOLD) hemodynamic response occurs over very slow timescales of several seconds (~0.1 Hz), and is typically sampled at a rate of around 1 Hz or less [2] – orders of magnitude slower than the millisecond timescales of underlying neural activity (>100 Hz). Moreover, fMRI measurements are noisy: sources of noise include scanner noise, head movement artifacts and physiological artifacts that systematically confound the fMRI signal [3]. Thus, fMRI BOLD activity reflects an indirect, noisy measure of neural activity, mismatched to neural timescales by several orders of magnitude. Previous studies have linked the BOLD response to underlying neurophysiological signals at various

---

∗Corresponding author

timescales [4–6], and have shown that fMRI data can be used to directly measure slow (<1 Hz) and infra-slow (<0.1 Hz) fluctuations in brain activity [7–10].

Here, we examined whether slow and infra-slow dynamics, as measured with slowly sampled fMRI data, are relevant for cognitive function, employing a novel combination of denoising, dimensionality reduction and machine learning. We applied Gaussian Process Factor Analysis (GPFA) – a recent technique for concurrent smoothing and dimensionality reduction of noisy data [11] – to fMRI signals sampled at ~0.3-1.4 Hz (scanner repetition time, TR of 0.72-3.3 s). GPFA enables the discovery of hidden (latent) dimensions in the data [11], with each latent dimension comprising a collection of channels that exhibit shared variance in their activity at a specific timescale. In our case, this approach enabled us to identify functionally connected networks of brain regions, whose dynamics evolve at diverse, characteristic timescales.

We also asked if slow fluctuations in fMRI latent dimensions discovered by GPFA would be sufficiently informative about task-specific cognitive states. While previous studies have examined slow, spontaneous dynamics in the resting state (e.g. [12]), we examined these dynamics with GPFA applied to both task-based and resting state fMRI scans, from 1000 healthy human participants drawn from the human connectome project (HCP) database [13]. These fMRI scans were acquired as subjects performed each of seven cognitive tasks – including tasks of working memory, story comprehension, gambling, and the like (Supporting Information Table S2) – or in the resting state. We discovered that GPFA activity in latent dimensions with characteristic slow and infra-slow timescales was remarkably synchronized across individuals' brains. Moreover, these slow GPFA dynamics could be used to predict with high accuracies (typically over 95%), task-specific cognitive states. Finally, applying GPFA to resting state fMRI data from patients with mild cognitive impairment (MCI), we show that infra-slow oscillations in these latent dimensions sufficed to distinguish, significantly above chance, MCI patients who later developed AD from those who did not. We propose that slow and infra-slow fMRI dynamics may be relevant for a complete understanding of cognitive function, in health and in disease.

## 2 Identifying latent fMRI dimensions with slow, characteristic timescales

GPFA extracts smooth, low-dimensional latent trajectories from noisy, high-dimensional time series [11]. Briefly, GPFA first performs dimensionality reduction with conventional factor analysis (FA) to identify an initial set of latent dimensions. This is followed by learning key parameters that specify the temporal autocorrelation of the latent dimensions (Fig. 1A). The parameters of the GPFA model are learned with an expectation-maximization algorithm. GPFA, therefore, provides a method for simultaneously performing smoothing and dimensionality reduction on the fMRI data. As with FA, GPFA incorporates an explicit noise model, which in the case of fMRI data enables estimating independent noise variances across different brain regions. Details regarding the model are described in [11] and in the Supporting Information (section 1).

We analyzed fMRI data from 1000 human participants (529 females, typical age range: 22-35 years, 10 subjects of age >36 years) drawn from the Human Connectome Project (HCP) database (total of 8000 scans). These scans were acquired when subjects were performing one of seven tasks (Supporting Information Table S2) or were resting inside the scanner (resting state); scanning and acquisition protocols are described elsewhere [13]. We used minimally preprocessed scans available from the HCP database; such preprocessing minimizes noise due to extraneous sources, such as scanner-related distortions or head movements inside the scanner [14]. Typically, each fMRI scan comprises a 4-dimensional dataset ($91 \times 109 \times 91$ voxels in space, and $176 - 405$ time points). As a first step, we parcellated the brain into 264 functionally-defined regions of interest (ROIs) using the Power et al [15] parcellation, which groups functionally related voxels into non-overlapping ROIs.

GPFA was run on the parcellated fMRI time-series to extract task-specific latent dimensions, and their associated trajectories. For this, we employed scans from 100 subjects (IDs in red in Supporting Information, Table S1), and the first 100 timepoints from each ROI (Fig. 1D, break in time axis). Before applying GPFA, we confirmed that a majority of these time series (~80%) satisfied Gaussianity assumptions as assessed by the Lilliefors test for normality. These timeseries were z-scored for each ROI separately and provided as input to the GPFA algorithm. Each subject was treated as a distinct experimental trial, and the GPFA model was trained with 4-fold cross validation (parameters trained on 75 subjects, and tested on 25 subjects, in each fold). Prediction errors were computed, with the test subjects' data, across a range of latent dimensions (5-100). The optimal number of latent dimensions

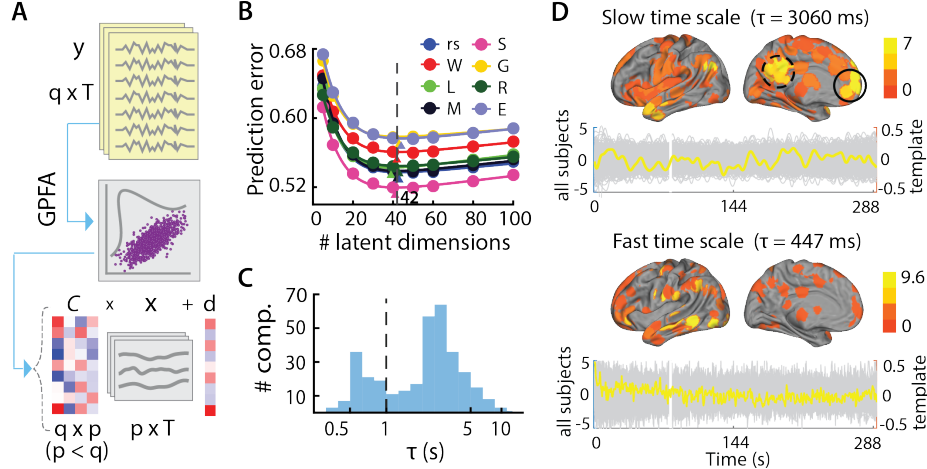

Figure 1: **Extracting latent dimensions at slow timescales with Gaussian Process Factor Analysis (GPFA). A** Schematic depicting the application of GPFA to fMRI time series data for concurrent smoothing and dimensionality reduction (see text for details). **B** Variation of prediction error with the number of latent GPFA dimensions. Lines: curve fits; triangles: minima for each curve. Dashed vertical line: Number of latent dimensions corresponding to minimum prediction error across tasks (42). Colors: Prediction errors for different tasks. *rs*: resting; *W*: working memory; *L*: language; *M*: motor; *S*: social cognition; *G*: gambling; *R*: relational processing; *E*: emotion processing. **C** Characteristic timescale ($\tau$) distribution; data pooled across tasks. Dashed vertical line: Threshold corresponding to slow (<1 Hz) timescale. **D** Representative spatial maps (column of $C$ matrix in panel **A**) for a slow timescale dimension ($\tau = 3060$ ms; top) and fast timescale dimension ($\tau = 447$ ms; bottom). The slow timescale dimension shows a distributed spatial map characteristic of default mode network (DMN), a canonical resting state network comprising the medial prefrontal cortex (solid circle) and posterior cingulate cortex (dashed circle). For clarity, the spatial maps depict only positive values of $C$. In each row, the left and right images show, respectively, the lateral and medial views of the brain's left hemisphere. Time series for the latent dimensions are shown below the corresponding maps. Gray: time series for individual subjects; yellow: average time series.

was then determined as the one that minimized prediction error, using a leave-region-out approach (further elaborated in the Supporting Information, section 1).

This approach revealed a clear minimum of the prediction error, corresponding to optimal reduced dimensionality for the fMRI data (Fig. 1B) for each of the 7 tasks and resting state: the minimum number of latent dimensions ranged from 39-44, indicating a nearly 6-fold reduction in data dimensionality. For further analysis, we determined a common number of optimal latent dimensions ($u$=42) across tasks, by minimizing the overall prediction error (Supporting Information, section 1). GPFA was then run, again, for each task with this common number of latent dimensions.

Each latent dimension $i$ ($= 1, ..., p$) estimated by GPFA can be described by the following quantities (Fig. 1A): i) a time series given by $\boldsymbol{x}_{i,:} = [x_{i,1}, \ x_{i,2}, \ \ldots, \ x_{i,T}]$; ii) the $i^{th}$ column of the mapping matrix $\boldsymbol{C}$, which specifies the contribution of each of the 264 brain regions, to latent dimension $i$, which we term the "spatial map" associated with the $i$th latent dimension. This map may be interpreted as a group (or network) of brain regions exhibiting shared latent dynamics governed by $\boldsymbol{x}_i$; and iii) a characteristic timescale for that latent dimension, $\tau_i$.

Across all tasks GPFA latent dimensions exhibited a bimodal distribution of timescales (Supporting Information, Fig. S2), with nearly 75% of dimensions exhibiting timescales slower than 1000 ms (1 Hz) (Fig. 1C, timescales pooled across tasks). Figure 1D shows a representative set of latent dimensions at slow and fast timescales obtained from the resting state scans. The slow timescale dimension (Figure 1D, top) exhibited a characteristic timescale of $\tau$=3060 ms. The spatial map for this dimension revealed a pattern characteristic of the default mode network (DMN), a widely-documented resting-state brain network comprising the medial prefrontal cortex, posterior cingulate cortex, precuneus and angular gyrus [16]. On the other hand, the fast timescale dimension (Figure 1D, bottom) exhibited a characteristic timescale of $\tau$=447 ms, which was faster than the sampling

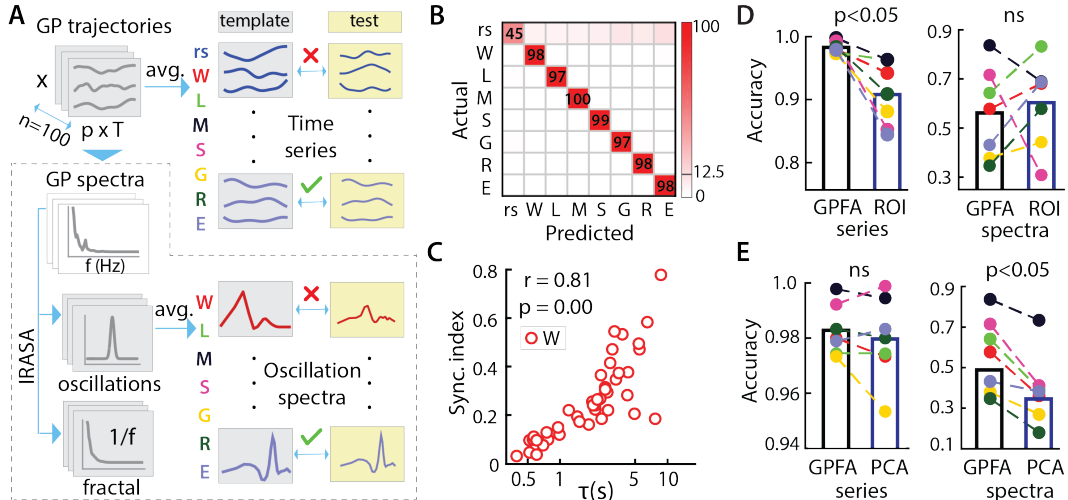

Figure 2: **Classifying task-specific cognitive states with GPFA latent dimensions. A** Schematic of classification with template pattern matching, based on latent trajectories (top) or oscillatory power (bottom) in GPFA latent dimensions (see text for details). **B** Confusion matrix for an 8-way classification showing the proportion of task (or resting) scans that were correctly classified or misclassified, using a template matching approach based on latent time series (see text for details). Chance: 12.5%. **C** Synchronization index (y-axis), measuring the average correlation among latent time series across subjects for each latent dimension, as a function of the characteristic timescale for that dimension (x-axis), for the working memory task (all tasks shown in SI Fig. S2). **D** (left panel) Comparison of accuracies for classification based on GPFA latent time series (left bar and points) and that based on 42-ROI time series (right bar and points) for each of the 7 tasks. Bars: Mean accuracies; (right panel) Same as in left panel but using ROI oscillation spectra. Color conventions: same as in Fig. 1B. **E** Same as in panel D but comparison with classification accuracies based on PCA dimensions: time series (left panel) and oscillation spectra (right panel).

frequency of the fMRI timeseries (1.4 Hz). GPFA latents with such fast timescales likely reflect artifacts, estimated from fits to residual noise after accounting for slow latents (see also Supporting Information, section 2).

## 3 Classifying task-specific cognitive states with slow latent dynamics

Next, we tested whether these latent dimensions carried information about task-specific cognitive states: Could examining these latent dimensions for each individual subject permit classifying the cognitive task that that subject was performing inside the scanner? For this, we designated the spatial maps computed by running GPFA on 100 subjects' data as "template" maps (42 per task scan) and then averaged the latent trajectories across these subjects as "template" trajectories (also 42 per task scan, each corresponding to one template map).

We employed these "template" latent dimensions to classify task-specific cognitive states for the remaining 900 subjects ("test" data; Supporting Information, Algorithm S1). Briefly, the time series $x(t)$ for some task scan $s$, for each subject in the test dataset, was projected onto the template maps of each of the eight tasks. This procedure generated a latent time series for each test subject's scan $s$, and was repeated with template maps from all eight scans. Following this the correlation between each test subject's projected (latent GPFA) trajectories and the template trajectories for each scan was computed, and summed across components, to get an overall similarity score. The template scan with the maximum overall similarity score with the test subject's scan, was assigned as the predicted task label for the test subject's scan. This was repeated for each scan, $s$=1-8, for each of the 900 test subjects, and all scans were assigned a specific label.

This approach (Fig. 2A) provided superlative accuracies for classifying among the seven different task states (confusion matrix; Fig. 2B). Median accuracy was 97.8%, and accuracies ranged from 96.7%-99.8%; all accuracies were significantly above chance (permutation test, $p < 0.001$). A clear exception was the resting state scan, which was often misclassified as a task scan. Nevertheless, this was an expected outcome because resting state data are not expected to be time-locked across subjects

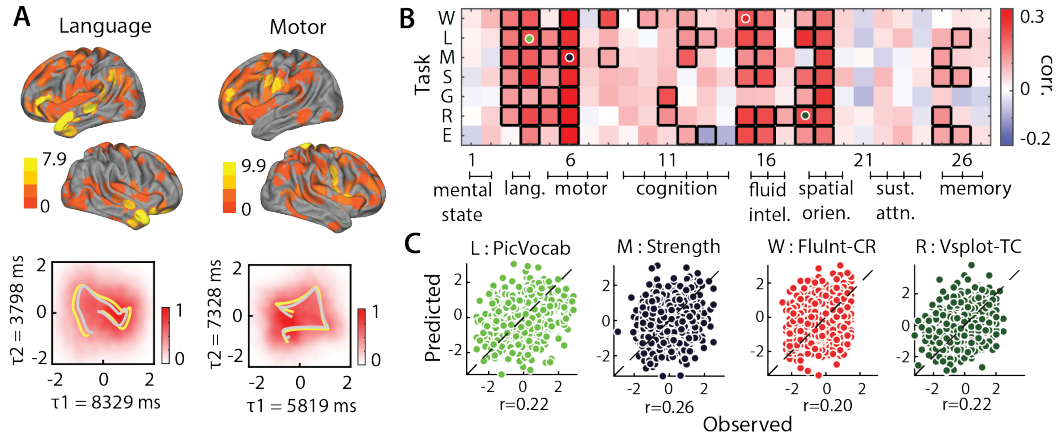

Figure 3: **Predicting behavioral scores with connectivity among GPFA latents. A** (Top) Spatial map of the most representative latent dimension for the language (left) and motor (right) tasks. The lateral views of the left and right hemispheres are shown for positive values of $C$ only. Maps for all tasks are shown in SI Fig. S1. (Bottom) Trajectories showing the joint activity for the most representative (x-axis) and second most representative (y-axis) latent dimensions for the corresponding task. Yellow and gray traces show average trajectories of template and test data, respectively. Red shading: normalized distribution of occupancy of the joint activity in the GPFA space spanned by these latent dimensions (n=900 subjects). Timescales corresponding to each dimension are marked along the respective axes. **B** Behavior scores across a range of cognitive tests (columns; refer SI Table S3) predicted based on GPFA connectivity in each task (row). Black outlined squares: significant predictions at the p<0.01 level with Benjamini-Hochberg correction for multiple comparisons. **C** Representative correlations of observed and predicted behavioral scores; all values were z-scored before plotting. (From left to right) picture-vocabulary test score based on the language task connectivity, strength score based on motor task connectivity, fluid intelligence sub-score based on working memory task connectivity and spatial orientation score based on relational task connectivity.

and scans. Therefore, estimating temporal correlations across latent GPFA dimensions should not be able to meaningfully identify the resting state. On the other hand, because the tasks performed by subjects all followed the same time-course, latent dimensions exhibited sufficient temporal structure – which was common and coordinated across subjects – to enable accurate classification. We repeated the classification limiting the time series to only those GP dimensions with slow timescales (<1Hz) and found comparably superlative accuracies across the seven tasks (median: 98% , range: 96.5% - 99.5% ). In addition, classification analysis following GPFA with a different (90-node) functional parcellation [17] also produced similar classification accuracies (Supporting Information, section 2, SI Fig. S3B), indicating that these results were not unique to a specific parcellation. Moreover, GPFA time series could distinguish, with superlative accuracies (range: 81%-99%) distinct sub-tasks within each task performed by the subjects inside the scanner (SI, section 2, Fig. S3D).

Next, we asked which spatial maps in the template data, were most representative of each task scan. For this we identified, for each task scan, the template dimension whose time series exhibited the highest correlation with the latent dimensions of the test dataset, for the corresponding scan (average correlation across n=900 test subjects). In other words, these dimensions represent brain networks which exhibited the most consistent dynamics (across subjects) for each task. Distinct and meaningful structure emerged in the spatial maps for each task. For instance, the language task showed clearly lateralized activation in the left superior temporal cortex, which includes the auditory cortex and language processing regions (Fig. 3A top left). The motor task, on the other hand, showed bilateral activation in the precentral gyrus, the location of the motor cortex (Fig. 3A top right). Other tasks, and the associated spatial maps, are shown in the Supporting Information (Fig. S1). Notably, we observed a strongly right hemisphere lateralized map for the gambling task (Fig. S1D), consistent with the findings of a recent study which showed that right lateralized brain activity was predictive of risk-taking behaviors [18].

We observed highly synchronized dynamics in the latent dimensions across subjects for each task (Fig. S1, second row). The average two dimensional latent trajectory for the template subjects (Fig. 3A, bottom, and Fig. S1, third row, yellow traces), were closely aligned with the average

trajectory for the test subjects (Fig. 3A, bottom, and Fig. S1, third row, gray traces), providing clear evidence for task-specific inter-subject synchronization of brain dynamics (see Discussion). We examined the degree of inter-subject synchronization, with a view to distinguishing task-relevant latent dimensions from task-irrelevant ones. We hypothesized that the strength of inter-subject synchronization within each latent dimension would index the task-relevance of that dimension. For example, artifacts, such as scanner noise or physiological noise, are unlikely to be synchronized across subjects, whereas dimensions strongly locked to salient aspects of the task would be highly synchronized. To quantify the degree of inter-subject synchronization for each latent dimension, we computed a "synchronization index": the average correlation between the template latent time series and the projected latent time series of each of the test subjects. A higher synchronization index indicates a higher degree of phase locking (or synchronization) of the test subjects' latent time series with the template time series. Slower timescale dimensions exhibited systematically stronger synchronization indices, as compared to faster dimensions; Fig. 2C shows this trend for working memory task (all tasks shown in SI Fig. S2). These results are consistent with the proposal that, across tasks, faster ($>$1 Hz) timescale dimensions reflected scanner artifact and noise, and slower ($<$1 Hz) timescale dimensions reflected task-relevant brain processes. They also suggest the potential utility of GPFA for denoising fMRI time series data (see Discussion).

We also tested if the power spectrum of GPFA latent dimensions contained sufficient information to distinguish among the tasks. For this, we first estimated the power spectrum of each dimension for each task using multitaper spectral estimation [19]. Next, we estimated and subtracted out the broadband (fractal or 1/f) component, using the IRASA (irregular-resampling auto-spectral analysis) method [20], to retain specifically the oscillatory component of the spectrum up to one quarter of the sampling rate (0.35 Hz). These oscillation power spectra were averaged across the template subjects (n=100) to form a "template" spectrum for each task and latent dimension (SI Fig. S4). As before, to classify each task scan of the "test" subjects' (n=900) data, we projected their timeseries based on the template spatial maps, computed the oscillatory spectra and correlated these with the oscillatory spectra of each template to identify the best matching template (Fig. 2A). Again, we discovered above-chance accuracies for task classification based on oscillation spectra: accuracies ranged from 34.7%-83.7% across the 7 tasks, with a median accuracy of 57.8% (p$<$0.01, permutation test).

We also performed additional control analyses to test if these results were specific to GPFA, or could be achieved with other dimensionality reduction approaches; these are described in the Supporting Information (section 2). Briefly, we reduced dimensionality either by selecting a subset of regions (ROIs) based on their activity correlation with the fMRI task timeseries, or with principal components analysis (PCA). In each case we compared the accuracy of task classification based on GPFA features – time series or oscillation spectra – with the accuracies obtained with these other approaches. Both GPFA latent spectra and time series provided significantly higher accuracies in classifying task-specific cognitive states, compared with at least one of the features in each of the other approaches (p$<$0.05, Wilcoxon signed rank text, Fig. 2D and Fig. 2E; details in SI section 2) .

We asked if, in addition to being able to identify task-specific cognitive states, GPFA latent dynamics would also be relevant as a marker of cognitive *traits*. We computed the functional connectivity between every pair of GPFA latent dimensions, based on partial correlations, for each subject. With these functional connectivity matrices as features, we sought to predict inter-individual differences in 27 cognitive scores acquired outside of the scanning session [21] (SI Table S3), using connectome-based predictive modeling (CPM; [22]). Behavioral scores were selected from Alertness, Cognition and Motor categories (HCP Data Dictionary), with the goal of avoiding redundant scores (e.g. sensitivity and specificity were included, but not also true and false positive rates). All scores were selected "blind" to (without *a priori* knowledge of) the results of these prediction analyses. Scores were predicted with 10-fold cross validation: by estimating the CPM model on a training fold with nine-tenths of the data while predicting scores on each left-out "test" fold (one-tenth of the data), in turn.

Many scores could be predicted significantly and almost universally across tasks (Fig. 3B, black squares; p$<$0.01 with Benjamini-Hochberg correction for multiple comparisons). These included scores of motor performance, fluid intelligence, linguistic ability and spatial orientation (Fig. 3B-C). Although the proportion of explained variance was comparatively low (mean $r$=0.16, range: 0.08-0.34), this range of correlations were similar to that observed in previous studies employing functional connectivity features for behavioral score predictions [23, 24]). On the other hand, scores associated with sustained attention, mental state or memory were predicted well with only some tasks or not at all.

We speculate that these differences may arise from the degree of mismatch between fMRI timescales and characteristic timescales for behavioral tasks used to measure these scores: Tasks engaging neural processes at timescales matching fMRI timescales (e.g. language or fluid intelligence) were possibly better predicted than those engaging processes at much faster (e.g. attention) or much slower (e.g. mental state) timescales. Overall, the results suggest that connectivity estimated with slow, latent fMRI processes may be relevant for predicting traits across several cognitive domains.

## 4 Predicting cognitive decline with infra-slow latent dynamics

As a second, key application of our approach, we sought to test whether infra-slow brain dynamics could serve as markers of cognitive decline. For this, we obtained resting state brain imaging (fMRI) scans from the Alzheimers Disease Neuroimaging Initiative (ADNI) database (adni.loni.usc.edu). Among the large number of patient datasets in this database, we utilized data from a subset of patients with Mild Cognitive impairment (MCI). MCI patients typically exhibit symptoms associated with decline of memory, language or thinking, that are usually more pronounced than normally aging adults. Nevertheless, only some proportion of such MCI patients progress to develop severe forms of dementia, like Alzheimer's Dementia (AD), as assessed by standard clinical ratings (e.g. clinical dementia rating, CDR). We term those MCI patients who progressed to develop AD as "MCI converters" (MCIc) and those who did not as "MCI stable" (MCIs) (Fig. 4B). Our goal was to test if infra-slow brain dynamics, as estimated with GPFA on resting fMRI data, would enable classifying MCIc from MCIs patients.

We analyzed resting state fMRI data for n=23 MCIc (age:72.7 $\pm$ 6.9 yrs, 11 females) and n=72 MCIs patients (age:71.6 $\pm$ 6.8 yrs, 32 females). For MCIs patients, scans were included only if the patient remained stably diagnosed as MCI for at least two years. For MCIc patients, scans were typically acquired 6 to 36 months (median: 12 months) prior to conversion to AD. One MCIc subject was excluded due to corrupted brain imaging data, so that data from a total of n=94 patients was analyzed. A standard pipeline was used to preprocess the scans and parcellate the brain into 264 regions, based on the Power et al. parcellation [15]. We then extracted latent dimensions by applying GPFA to parcellated fMRI time series, concatenated across all subjects (MCIc and MCIs). As before, we obtained an optimal number of latent dimensions ($u$=77) with a prediction error minimization approach. Because only resting state fMRI data was available from MCI patients no useful information could be obtained by examining correlations between different patients' latent trajectories. This is because brain fluctuations in the resting state are uncontrolled, in the absence of external task events. Thus, we sought features in the resting state GPFA dimensions that did not require consistent phase relationships between latent time series across subjects.

First, we computed partial correlations (PC) and lagged covariance (LC) between every pair of latent dimensions for each subject (Supporting Information, Section 4). These features are indicative of instantaneous or lagged connectivity across brain regions for each subject [25], and do not require activity to be synchronized across subjects. We then trained a linear classifier based on support vector machines (SVM) with PC and LC features as input, and distinct labels for the MCIc and MCIs patients, using Matlab's *fitlinear* function (Fig. 4A). As the numbers of selected MCIc scans (n=22) and MCIs scans (n=72) were substantially different, a balanced classes approach was used prior to classification to prevent classifier bias (Supporting Information, section 4).

Classification accuracy based on PC connectivity features was 61.2% whereas that based on LC connectivity features was 71.0%. Combining the two sets of features provided a marginally lower classification accuracy of 63.6%; each of these accuracies were significantly above chance as assessed with a permutation test based on shuffling labels (p<0.05; Supporting Information, section 4). To find a minimal set of features that provided the highest cross-validation accuracy (area-under-the-curve or AUC) we applied recursive feature elimination (RFE) including both PC and LC features [26] (Supporting Information, section 4). RFE retained only 15% (1332) features of the 8855 PC+LC features, and these few features sufficed to achieve a comparatively high classification accuracy of 73.6%. The top ranked three features, with the highest SVM beta weights, were all PC connections: the highest beta weights occurred for a PC connection between a predominantly left lateralized occipito-parietal map and a bilateral temporal map (Fig. 4C). The next two beta weights occurred for connections involving the fronto-insular cortex, and the anterior cingulate cortex (SI Fig. S5). These maps overlap strongly with the fronto-parietal "attention" network, and a cingulo-opercular "salience" network, conventionally implicated in attention and executive control, respectively [27].

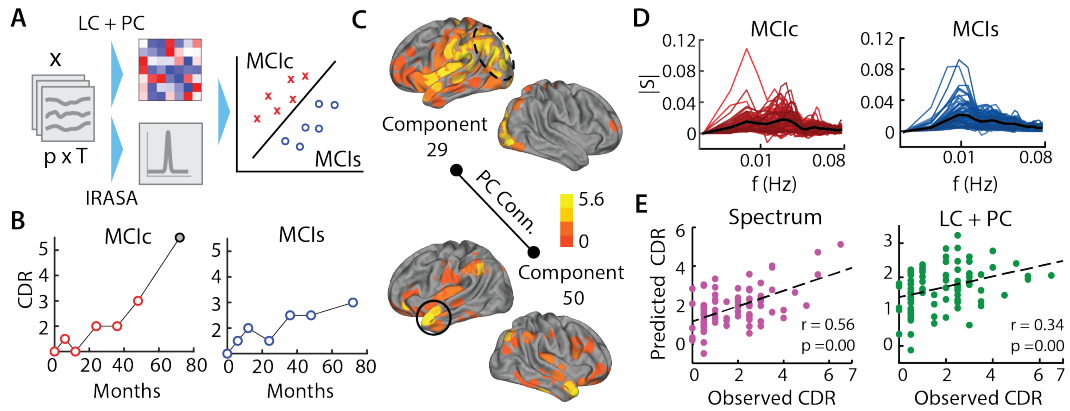

Figure 4: **Predicting cognitive decline with GPFA latent dynamics.** **A** Schematic of classification between MCI-converter and MCI-stable patients based on resting state connectivity and oscillatory dynamics (see text for details). **B** Progression of clinical dementia rating (CDR, sum of boxes) score for a representative MCI-converter (left) and an MCI-stable (right) patient across visits (months). Black data point in left graph denotes diagnosis of Alzheimer's Dementia (AD) onset. **C** Spatial maps of the most discriminative GPFA-latent connection for the MCIc versus MCIs classification (partial correlation between GP components 29 and 50). Dashed oval: occipito-parietal map. Solid circle: temporal map. **D** Oscillation power spectra across all 77 latent dimensions for MCIc patients (left) and MCIs patients (right). Black lines: average spectra across dimensions, show marked differences between MCIc and MCIs patients. **E** Observed CDR (sum-of-boxes) scores versus scores predicted with oscillation spectral features (left panel) and lagged covariance and partial correlations (right panel) of GPFA trajectories. Dashed line: linear fit.

Second, we tested if the power spectrum of the latent dimensions sufficed to classify between MCIc and MCIs subjects. As before, we computed the power spectrum for each component and subject and removed the broadband (1/f) component to retain only the oscillatory component (with IRASA). We then used these oscillatory power spectra (Fig. 4D) as features in an SVM classifier using a balanced classes approach, as discussed above. In this case, due to the slower sampling rate (longer repeat time) of the fMRI scan (0.3 Hz) compared to the HCP data (1.4 Hz), we included for the classification only frequencies up to ~0.08 Hz (infra-slow band), sampled uniformly at 32 values (total of 2464 features). Classification accuracy based on the SVM was 65.1%, and significantly above chance, as assessed by a permutation test (p<0.05). As before, following RFE, we discovered that, on average, only 28% (681/2464) features were retained whereas accuracy following RFE improved marginally to 67.2%.

Finally, we evaluated the ability of slow latent dynamics to predict a graded measure of cognitive decline. We predicted individual patients' CDR-SOB (CDR-sum of boxes) scores that measure severity of dementia [28], based on the resting fMRI scan obtained in the same visit. We fit a regression model with an L1 penalty (Matlab's *lasso* function) to predict CDR-SOB scores, separately, with power spectral features and with LC and PC features, with a k-fold cross validation approach (k=2, 4, 23, 46, 92; Supporting Information, section 4). The accuracy of prediction increased steadily with more folds, perhaps because of the small sample size and consequent bias with limited training data for small k. Leave-one-out cross-validation provided the most accurate predictions with power spectral features yielding a strong correlation (r=0.56, p<0.001, Fig. 4E left) between predicted and observed scores. LC and PC features yielded a marginally lower, but significant, prediction (r=0.34, p<0.001, Fig. 4E right).

Taken together, these analyses indicate that slow and infra-slow fluctuations in resting state fMRI data are reliable biomarkers that distinguish between MCI patients who converted to AD versus those who did not. Moreover, the results suggested compromised function in putative attention-related brain networks that could be indicative of progression to AD. Finally, slow dynamics in these networks predicted inter-individual differences in cognitive decline, as assessed by CDR-SOB scores.

# 5    Discussion

Slow (0.1-1 Hz) and infra-slow ($<0.1$ Hz) fluctuations in spontaneously generated electrical activity of the brain have been widely observed with various recording modalities, including local field potential microelectrode recordings [29], optical recordings [30], electrocorticography [31, 32] and electro- and magnetoencephalography [33]. Here, we examined whether such slow fluctuations, as estimated with GPFA applied on fMRI data, would be informative for understanding cognitive processes in healthy brains, and cognitive decline in diseased brains.

Applying GPFA to fMRI data revealed several key insights. First, GPFA reduced the dimensionality of the data by 3-6 fold, indicating significant redundancy in the data. This significant reduction in dimensionality is a consequence of the shared variance of the fMRI BOLD response across multiple regions. This shared variance could be exploited to produce sparse representations of fMRI brain activity. Moreover, it is possible that an even higher reduction in dimensionality could be obtained with GPFA applied to voxel-level fMRI data. At present, computational costs associated with scaling up GPFA to the voxel level prohibit such an application, and highlight the need for developing more efficient algorithms for voxel-level GPFA. Second, GPFA provides a key advantage over other dimensionality reduction techniques, like independent components analysis (ICA) [34], in that GPFA directly estimates latent dimensions with distinct, characteristic timescales. This feature enabled us to show that GP dimensions with fast timescale dynamics ($>1$ Hz) were not synchronized across subjects, and likely reflected artifactual processes (e.g. high frequency scanner noise). Prior knowledge about the timescales of scanner or physiological artifacts could be directly incorporated into the EM algorithm when estimating GPFA parameters, thereby permitting concurrent denoising of fMRI data.

On the other hand, GPFA latent dimensions with comparatively slower timescale dynamics ($<1$ Hz) were strongly synchronized across subjects (Fig. 2C, SI Fig. S2). Previous studies have reported inter-subject synchronization of cortical responses, at multiple different timescales. Studies that used either fMRI [35] or MEG [36] to image brain activity showed that cortical activity is highly synchronized across subjects experiencing naturalistic stimuli, either visual (e.g. movies) or auditory (e.g. music), particularly when salient events occur in the stimulus stream [37, 38]. A related study with task-based fMRI identified brain networks that showed consistent synchrony across the subject population[39]. Moreover, a key advantage of using GPFA over previous approaches is that GPFA provides considerable flexibility with identifying a variety of brain network dynamics. Alternative GP covariance kernels, such as a periodic kernel or a linear kernel [40] can help identify periodic fluctuations or drifts, respectively, in the fMRI data, which could then be matched with relevant task covariates of interest, or factored out as noisy confounds. For example, some types of motion artifacts, arising due to overt behavioral responses in the scanner, could be synchronized across subjects, and these could be identified and denoised with GPFA.

Our results suggest that BOLD dynamics that occur at infra-slow (~0.1Hz) timescales may serve as markers for cognitive function in health, and, cognitive decline, in disease. Understanding the link between these BOLD dynamics and dynamics of underlying neural processes remains an important open question. Despite this caveat, our approach is relevant for developing non-invasive, imaging-based biomarkers for early detection of cognitive impairment. For example, changes in slow and infra-slow BOLD dynamics can be measured longitudinally, over several years, as a potential biomarker for predicting the onset and trajectory of cognitive decline in AD, and could form the basis for early intervention before severe behavioral symptoms of cognitive decline manifest.

# 6    Acknowledgements

We wish to thank Byron Yu and Abhijit Chinchani for guidance with GPFA analysis. This research was funded by a Wellcome Trust-Department of Biotechnology India Alliance Intermediate fellowship [IA/I/15/2/502089], a Science and Engineering Research Board Early Career award [ECR/2016/000403], a Pratiksha Trust Young Investigator award, a Department of Biotechnology-Indian Institute of Science Partnership Program grant, a Sonata Software grant and a Tata Trusts grant (to DS). We would like to thank the Human Connectome Project (HCP) and Alzheimers Disease Neuroimaging Initiative (ADNI) for access to fMRI data; details regarding their funding sources are available in their respective websites (http://www.humanconnectomeproject.org/; https://adni.loni.usc.edu).

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
