[Supplementary Material]

# Supporting Information:
# Infra-slow brain dynamics as a marker for cognitive function and decline

**Shagun Ajmera**
Centre for Neuroscience
Indian Institute of Science
Bangalore
ajmerashagun@gmail.com

**Shreya Rajagopal**
Centre for Neuroscience
Indian Institute of Science
Bangalore
shreyakr96@gmail.com

**Razi Ur Rehman**
Computer Science and Automation
Indian Institute of Science
Bangalore
razirmp@gmail.com

**Devarajan Sridharan**$^*$
Centre for Neuroscience &
Computer Science and Automation
Indian Institute of Science
Bangalore
sridhar@iisc.ac.in

## 1 Gaussian Process Factor Analysis (GPFA)

**Description of the technique**. We define $\boldsymbol{y}_{:,t} \in \mathbb{R}^{q \times 1}$ to be the high-dimensional fMRI data (observed data) with $q$ voxels or regions of interest (ROIs). The fMRI time series is sampled at time points $t = 1, ..., T$. GPFA extracts a set of low-dimensional latent states, $\boldsymbol{x}_{:,t} \in \mathbb{R}^{p \times 1}$ $(p < q)$, by defining a linear-Gaussian relationship between the observations $\boldsymbol{y}_{:,t}$ and latent states $\boldsymbol{x}_{:,t}$ as: $\boldsymbol{y}_{:,t} | \boldsymbol{x}_{:,t} \sim \mathcal{N}(\boldsymbol{C}\boldsymbol{x}_{:,t} + \boldsymbol{d}, \boldsymbol{R})$ where $\boldsymbol{C} \in \mathbb{R}^{q \times p}$, represents the mapping between the observed (fMRI) data and the latent dimensions, $\boldsymbol{d} \in \mathbb{R}^{q \times 1}$ and $\boldsymbol{R} \in \mathbb{R}^{q \times q}$ represent the mean of the observed data, and the independent variances (of putative noise) across the ROIs (or voxels), respectively, and $\mathcal{N}$ denotes the normal distribution. Thus, the presence of independent noise across ROIs can be factored into $\boldsymbol{R}$, whereas correlated noise across ROIs are factored by GPFA into a separate latent dimension. A key difference between conventional dimensionality reduction techniques, like PCA, and GPFA is that the latter specifies each latent dimension to be a Gaussian process, with a particular form of the temporal (auto)covariance matrix. Thus, each latent dimension $i$ is given by: $\boldsymbol{x}_{i,:} \sim \mathcal{N}(0, \boldsymbol{K}_i)$ where $\boldsymbol{K}_i \in \mathbb{R}^{T \times T}$ is the temporal covariance matrix for the $i$th latent dimension. Here, we choose a form of the covariance that is conventionally employed to generate smooth latent trajectories: the squared exponential function. $\boldsymbol{K}_i(t_1, t_2) \propto \exp(-\frac{(t_1 - t_2)^2}{2\tau_i^2})$ where $\boldsymbol{K}_i(t_1, t_2)$ denotes the covariance between time points $t_1$ and $t_2$ of latent dimension $i$ and $t_1, t_2 = 1, ..., T$. The parameters of the GPFA model, viz., $\theta = \{\boldsymbol{C}, \boldsymbol{d}, \boldsymbol{R}, \tau_1, ..., \tau_p\}$ are learned using an expectation maximization (EM) algorithm [1]. Because the variance of BOLD data in each region could change with the mean, each of the BOLD time series were variance normalized by z-scoring (mean subtracted and divided by standard deviation) before applying GPFA.

**Estimating the number of latent GPFA dimensions**. To identify the optimal number of GPFA latent dimensions $p$ for the fMRI data, we employed an approach similar to the leave-neuron-out approach of Yu et al (2009) [1]. The "template" subjects' data were divided into four folds, such that each fold comprised 25 subjects' data. GPFA was run on all-but-one folds of the data (training data; $[\mathbf{Y}_{\text{tr}}]_{q \times 75 \cdot T}$), including data from all brain regions. Then the observed data from all-but-one regions of the left-out fold (test data; $[\mathbf{Y}_{\text{te}}]_{(q-1) \times 25 \cdot T}$) was projected using the mapping matrix estimated

---

$^*$Corresponding author

---

**Algorithm 1** Task classification based on GPFA features

---

**Input:**
task labels *task*, size $(1 \times 8)$
test scan data *test_scans*, size $(900 \times 8 \times q \times T)$
template trajectories $\boldsymbol{x}$, size $(8 \times p \times T)$
timepoints $T = 1 : 100$
parameters *GPFA_par*$\{\boldsymbol{C}, \boldsymbol{d}, \boldsymbol{R}, \boldsymbol{\tau}\}$
Initialize *ConfusionMatrix*=zeros$(8 \times 8)$
**for** subject $S$, task scan $s$ **do**
   $y = test\_scans(S, s, 1 : q, T)$
   Reset $\rho_{max}$, *predicted_label*
   **for** *task*$= 1$ **to** $8$ **do**
      $\hat{x}_{task} = estimateLatentState(y, GPFA\_par)$
      $\rho = \sum_{C=1}^{p} corr(\hat{x}_{task,C,:}, x_{task,C,:})$
      **if** $\rho > \rho_{max}$ **then**
         $\rho_{max} = \rho$
         *predicted_label* = *task*
      **end if**
   **end for**
   *ConfusionMatrix*($s$,*predicted_label*)++
**end for**

---

from the training data, including all-but-one rows, corresponding to the left-out region, $([\boldsymbol{C}_{\text{te}}]_{(q-1) \times p})$ to obtain a set of latent trajectories for the test data $([\mathbf{X}_{\text{te}}]_{p \times 25 \cdot T})$. Using these trajectories and the left out row of the mapping matrix $([\boldsymbol{C}_{\text{te}}]_{1 \times p})$, the data for the left-out region was reconstructed as $[\hat{\mathbf{Y}}_{\text{te}}]_{1 \times 25 \cdot T}$). RMS error was computed between this prediction and the observed regional time-series of the left-out region $([\mathbf{Y}_{\text{te}}]_{1 \times 25 \cdot T})$. This was repeated by leaving out each region in turn, and the summed prediction error was computed across all left out regions and subjects in the test data. The process was repeated, again, by assigning each fold of the data as the test set, in turn. The prediction error was computed for a range of number of latent dimensions $p = 5$ to $100$, in steps of $5$. This was done for each task separately, to obtain separate estimates of optimal dimensionality for each task scan set (Fig.1B, main text).

Because these estimates were closely similar, for ease of further analysis,we computed a common number of optimal latent dimensions $(u)$ that minimized the prediction error across all the tasks:

$$\min_{u \in [5,100]} \sum_{t \in \texttt{tasks}} [\text{E}^t(u) - \text{E}^t(u_{min}^t)]$$

where $\text{E}^t(u)$ is the prediction error with $u$ latent dimensions for task $t$ and $u_{\texttt{min}}^t$ is the number of latent dimensions that minimized the prediction error for task $t$. With fMRI data parcellated with the Power et al. (264 ROI) parcellation [2], this approach provided a common number of $u$=42 cross-validated data dimensions, indicating a 6-fold reduction in dimensionality of raw fMRI data, across tasks.

## 2 Classifying task-specific cognitive states in healthy subjects

**Permutation testing of classifier accuracies**. The classification accuracies mentioned in the main text were tested for significance, by shuffling the task labels of all scans of the 900 subjects in training stage (before correlating with templates). The labels were predicted the same way as described in Algorithm 1, and compared against the true labels. The labels were shuffled and classification repeated 1000 times to get a null distribution of classification accuracies. Significance level for the classification accuracies reflect the proportion of values in the null distribution that were greater than the actual classification accuracies computed based on GPFA features.

**Classification with alternative parcellations**. To test whether the superlative classification accuracies were specific to the Power et al [2] parcellation, we tested the GPFA time series based classification accuracies with an alternative parcellation. We employed the Shirer et al (90-node) parcellation – an alternative, widely used, function parcellation [3]. As before, we computed the

number of latent dimensions using a prediction error minimization approach outlined in Section 1 (above), and discovered an optimal dimensionality of $u$=17; again GPFA indicated a 5-6 fold reduction in data dimensionality consistent with estimates from the previous parcellation (SI Fig. S3A). All other procedures – including computing template trajectories, classification based on matching test scan trajectories to template trajectories, computing accuracies, and the like, followed the procedures outlined in the main text (Section 3). As before, we observed superlative accuracies for classifying the seven task states (confusion matrix; SI Fig. S3B). Median accuracy was 96.6%, and accuracies ranged from 89.2% - 99.3% across the seven tasks. All accuracies were significantly above chance (permutation test, p<0.001). Again, the clearest exception was the resting state scan, which was often confused with one of the other task scans. These results confirmed that the accuracies reported in the main text were not a consequence of a specific parcellation scheme.

**Classification among sub-tasks**. In the HCP database, tasks performed by subjects during scanning were not homogenous: rather, these occurred in interleaved blocks of distinct sub-tasks (Table S2, Supporting Information). For example, the working memory task comprised of a 0-back sub-task, and a more cognitively demanding 2-back sub-task; in the latter sub-tasks, items had to be held longer in working memory. Similarly, the motor task comprised of separate blocks of trials that involved movements of the hand, foot or tongue. We concatenated blocks of GPFA latent time-series corresponding to each sub-task together and tested if these time-series would suffice to classify between the sub-tasks within each task. Again, we obtained superlative classification accuracies ranging from 81% - 99% (p<0.001, permutation test). This suggested that latent dynamics estimated with GPFA were sufficient to discriminate finer grained differences in cognitive states among the sub-tasks of each task (SI Fig. S3D).

**Extracting oscillatory spectral features**. We also employed a recent technique for isolating the oscillatory component from the fractal component of the GPFA latent dimensions' spectra: Irregular-Resampling Auto-Spectral Analysis (or IRASA). Details regarding the technique can be found in [4]. Briefly, IRASA iteratively upsamples and downsamples the time series at irregular (non-integral) factors, and estimates the average spectrum across these different sampling factors. This provides an estimate of the fractal component of the spectrum; the intuition behind the approach is that the fractal component of the spectrum should be invariant to scaling up or scaling down the sampling interval. Then the fractal spectral estimate is subtracted out from the power spectrum of the original data to isolate the oscillatory (non-fractal) component of the spectrum. Because of this subtraction, the oscillatory power spectrum may occasionally contain negative values, as shown in SI Fig. S4.

Combining GPFA with IRASA enabled estimating slow timescale oscillatory processes with high specificity. Previous fMRI studies have typically examined slow and infra-slow fluctuations in fMRI data by filtering low-frequency activity from fMRI time series data, and quantifying the power of these low-frequency fluctuations with conventional spectral analysis approaches (e.g. FFTs, [5–7]). Nevertheless, such low frequency fluctuations may be contaminated with various artifacts of physiological, and non-physiological origin. Furthermore, because spectral power in most natural processes decays as 1/f, low-frequency power directly estimated from fMRI responses is likely dominated by non-oscillatory "fractal" content. IRASA overcame these shortcomings by enabling us to selectively quantify oscillatory power in the latent dimensions. We discovered that with infra-slow oscillatory power estimated with IRASA and GPFA, we could classify task specific cognitive states in healthy subjects, as well as the potential for conversion of MCI patients to AD.

**Comparison of GPFA classification accuracy with other dimensionality reduction approaches**. We performed two control analyses to test if these results were specific to GPFA, or could also be achieved with other dimensionality reduction approaches. First, we reduced data dimensionality by selecting a subset of the 264 parcellated ROIs, and performing task classification using their time series directly. For each task, we selected the top 42 ROIs (same number of latents as GPFA) based on the correlation of their time series with the underlying task structure (elaborated in the next paragraph). In this case classification accuracies for the 7 tasks ranged from 84.4% to 96.3% (median 90.9%), and were significantly worse than accuracies based on GPFA (signrank test, p=0.0156, Fig. 2D left, main text). Next, we computed these accuracies based on the spectral features from the same 42 ROIs. In this case, classification accuracies were not significantly different from those based on GPFA spectral features (median=68.1%, range=31.0−83.1%, p=0.58, Fig. 2D right, main text). Second, we compared classification accuracies based on dimensionality reduction with principal component analysis (PCA, Supporting Information, Section 3). In this case, we discovered no significant difference between task classification accuracies based on GPFA versus those based on

PCA time series (p=0.66, signrank test; Fig. 2E left, main text). In fact, our analysis revealed an empirical relationship between PCA and GPFA time series, once the dimensions of the latter were orthogonalized (Supporting Information, Section 3). However, classification accuracies based on PCA spectra were systematically worse than those based on GPFA spectra (p=0.0156; Fig. 2E right, main text). In summary, both GPFA time series and spectral features outperformed other dimensionality reduction approaches in terms of their accuracy with classifying task-specific cognitive states.

**Classification with ROI time series**. As one way to reduce dimensionality, we identified a subset of ROIs from the 264 node Power et al. parcellation, which were most representative for each task. For this, we concatenated the 100 template subjects' ROI series (the first 100 timepoints), and normalized their amplitudes by z-scoring. We then fit a separate multivariate linear regression model for each ROI series, with each sub-task time-series modeled as a box-car function, with a value of 1 at scans (time points) during the occurrence of each sub-task and 0 otherwise. For each of the 7 tasks, we ranked the ROIs based on the magnitudes (absolute values) of their regression coefficient ($\beta$), pooled across sub-tasks for that task. The top 42 ranked ROIs (identical with the number of GPFA components) were identified for each task. Then a pattern-matching based classification (same as Section 3 main text) was performed on the 900 test subjects using the 42-dimensional ROI trajectories. We performed this analysis for the 7 tasks only, excluding resting state, because resting state fluctuations are uncontrolled and not temporally coordinated across scans and subjects. We obtained inferior accuracies for ROI time series based classification compared to that with GPFA time series (Fig. 2D left, main text; see previous paragraph). We repeated this analysis with a 90 node Shirer et al. parcellation, using only the top 17 ROIs for classification. In this case also, the accuracies for the 7 tasks ranged from 59.8% to 94.7% (median 83.8%) (refer to SI Fig. S3C); again GFPA time series-based accuracies outperformed accuracies based on ROI time series.

**Predicting behavioral scores with GPFA connectivity**. We predicted individual subjects' behavior scores based on the functional connectivity, estimated with partial correlations [8] among the 42 GPFA components for each subject (total of 861 connection features). This analysis was limited to 871/1000 subjects for whom all of the 27 behavioral scores were available. The prediction was performed with the connectome-based predictive modeling approach [9]. Briefly, this approach involves a preliminary feature selection step in which the strength of each connection is correlated with behavioral scores, across subjects (861 univariate correlations for each behavioral score). Connections with significant (at the p<0.05 level) positive and negative correlations were identified and summed separately, to yield the overall strength of connections positively and negatively correlated respectively, with each behavioral score. Then a general linear model was fit with the summed positive and negative connection strengths as independent variables, and the behavioral score as the dependent variable. Using the regression coefficients for the model fit, the behavioral score of the test subjects was predicted based on their, respective, connectivity values. Predictions were assessed with 10-fold cross-validation such that both feature selection and regression model fitting was performed on nine-tenths of the data, and behavioral scores were predicted on the left out (one-tenth) of the data, with each fold of left out data being used in turn for the predictions.

**Potential artifacts captured by fast timescale latent dimensions**. Across tasks, latent dimensions whose timeseries were least correlated across subjects (with the lowest synchronization index) typically exhibited fast characteristic timescales (SI Fig. S2). These dimensions likely reflect potential artifacts such as high frequency scanner artifacts or physiological noise, rather than fast neural processes. To explore these fast timescale dimensions further, we examined the spatial maps of the five latent dimensions with the least synchronization index, for each task. SI Fig. S6 shows an exemplar dimension that occurred with a uniformly low synchronization index across all tasks. The spatial map for this dimension was nearly identical across tasks, and resembled high frequency spatial noise, with phase inverted across the hemispheres. We speculate that this dimension represents a scanner artifact arising from field inhomogeneities that occurred consistently across subjects and tasks. Although we applied GPFA to gray matter ROIs, GPFA applied to whole brain voxel-level fMRI data may be able to identify and denoise contributions from other noise sources, including CSF or white matter signals.

## 3   Comparison of GPFA with Principal Components Analysis (PCA)

We compared the classification accuracies obtained with GPFA dimensions against another conventional dimensionality reduction technique: PCA. To compare these accuracies, we adopted the

following procedure: First, we parcellated the data, as before with the Power et al (264 node) parcellation. Next, we ran PCA for each of the task scan and resting scan datasets from the "template" subjects (n=100) to identify a set of principal components unique to each scan. We then selected the top 42 components ranked based on their explained variance. This number was chosen to be identical with the number of GPFA components selected; the results remained similar, even if we chose the number of components based on 95% of cumulative explained variance (n=205-216 components, across tasks). As before, test subjects' data (n=900) were projected onto these principal components – every scan of the test subjects' data was projected into the PCA space corresponding to each template scan data. We then performed classification of task specific cognitive states, using temporal and spectral features of these PCA projected latent time series.

First, with a procedure identical to that outlined in Algorithm 1, we labeled the template scans based on the closest match (highest correlation) to the template scan time series. We observed that classification accuracies based on PCA time series ranged from 94.7%-99.9% (median 97.7%) across the seven tasks; as before, resting state classification accuracies were the least (47%). PCA and GPFA classification accuracies were not significantly different (p>0.05, signrank test; Fig. 2E left, main text). Second, we performed the same classification analysis using oscillatory spectral features of the PCA latent dimensions of the 7 tasks, again, using the same procedure as described in the main text for the GPFA latent dimensions. In this case, we observed that classification accuracies based on PCA spectra ranged from $18.7\% - 74.0\%$ (median 38.9%) across the seven tasks. Moreover, classification accuracies based on PCA spectra were significantly poorer than GPFA-spectra classification accuracies (p<0.05, Fig. 2E right, main text).

We sought to reconcile these differences in classification accuracies between PCA and GPFA. A key difference between PCA and GPFA is the following: PCA latent dimensions are orthogonal, by definition. On the other hand, GPFA dimensions are not constrained to be orthogonal: the columns of the mapping matrix $\boldsymbol{C}$ are not constrained to be orthogonal. Nevertheless, a "PCA-like" mapping can be obtained for GPFA components using an orthonormalization procedure; this process does not alter the GPFA model-fitting procedure but involves rotating the matrix $C$.

To implement this orthonormalization, we apply singular value decomposition [10] to the learned $\boldsymbol{C}$ [1]. This yields $\boldsymbol{C} = UDV'$, where $U \in \mathbb{R}^{q \times p}$ and $V \in \mathbb{R}^{p \times p}$ each have orthonormal columns and $D \in \mathbb{R}^{p \times p}$ is diagonal. We can then write:

$$\boldsymbol{y}_{:,t} = \boldsymbol{C}\boldsymbol{x}_{:,t} = U(DV\boldsymbol{x}_{:,t}) = U\tilde{\boldsymbol{x}}_{:,t}$$

where $\tilde{\boldsymbol{x}}_{:,t} = DV\boldsymbol{x}_{:,t} \in \mathbb{R}^{p \times 1}$ is the orthonormalized latent dimension at time point $t$, and is a linear transformation of $\boldsymbol{x}_{:,t}$. Since $U$ has orthonormal columns, we can now visualize the trajectories associated with $\tilde{\boldsymbol{x}}_{:,t}$ in an orthonormalized space. Specifically, because the elements of $\tilde{\boldsymbol{x}}_{:,t}$ (and the corresponding columns of $U$) are ordered by the amount of data covariance explained, these are directly analogous to PCA [1]. This provides a way to directly compare GPFA orthonormalized dimension trajectories with latent trajectories generated by PCA.

As with GPFA, we visualised the top two most representative latent dimensions of PCA and orthonormalized GPFA (SI Fig. S3E shows for Motor and Social cognition tasks). GPFA trajectories in this orthonormalized space, were remarkably similar to PCA trajectories; the mirror symmetry that occurred in some trajectories is a consequence of the fact that both PCA and GPFA dimensions are sign agnostic. This revealed a close empirical relationship between GPFA (orthonormalized) dimensions and PCA dimensions, indicating that both carry nearly identical temporal information. Because the orthonormalized dimension ($\tilde{\boldsymbol{x}}_{:,t}$) is a linearly transformed version of its non-orthonormalized counterpart $\tilde{\boldsymbol{x}}_{:,t}$, it is not surprising that classification accuracies based on the latent dimension time series were not significantly different between PCA and GPFA. On the other hand, unlike the non-orthonormalized GPFA dimensions, the orthonormalized dimensions reflected a mixture of autocovariance timescales and, therefore, no longer exhibited the smooth trajectories apparent in the GPFA dimensions (compare with SI Fig. S1C,E last row). Therefore, spectral content (that strongly depends on temporal autocovariance) was likely to be significantly different in the GPFA (non-orthogonalized) and PCA dimensions, which could explain the difference in classification accuracies between GPFA and PCA, based on spectral features.

# 4 Predicting cognitive decline in MCI patients

**Feature extraction**. In order to classify patients with mild cognitive impairment (MCI) who progressed to develop Alzheimer's dementia (MCIc), from those who did not (MCIs), we employed the following features. First, we computed zero-lag partial correlations among the GPFA latent dimensions, separately for each MCIc and MCIs patient; PC was computed using the *partialcorr* function in Matlab. With 77 latent dimensions, and because the PC matrix is symmetric, this feature space contained 2926 features comprising the lower triangular entries of the PC matrix; diagonal values (all 1-s) were excluded. Next we computed lagged (lag-1) covariance (LC) among the different GPFA dimensions. These were computed with the *mrdivide* function in Matlab, as $\mathbf{X}_1(\mathbf{X}_0)^+$, where $\mathbf{X}_0$ denotes the latent timeseries, $\mathbf{X}_1$ denotes the lag-1 latent time series, and $^+$ denotes a pseudoinverse operation. With 77 latent dimensions, and because the LC matrix is not (generally) symmetric, this feature space contained 5929 features comprising all entries of the LC matrix. Thus, the SVM classification and recursive feature elimination were performed in a feature space with 8855 dimensions.

**Distinguishing MCI-converters from MCI-stable patients**. We used a support vector machine (SVM) classifier, using the *fitclinear* function in Matlab, to distinguish MCIc from MCIs patients. From the sample of data analyzed (23 MCIc and 72 MCIs patients), one MCIc subject was excluded because of corrupted imaging data. Due to the imbalance of class labels in the data (now, 22 MCIc and 72 MCIs patients) a balanced classes approach, with an equal number of MCIc (n=22) and MCIs (n=22) subjects, was used prior to classification, to avoid classifier bias. Across 100 runs of SVM training and testing, 22 MCIs subjects pseudo randomly sampled from the pool of 72, as well as all 22 MCIc subjects were used. Each MCIs subject was sampled at least once every 4 runs. Therefore, across runs, the minority class (MCIc) was over sampled and the MCIs data were sampled so that each subjects' scan was sampled at least 25 times, with some subjects' scans randomly sampled with higher frequency. For leave-one-out cross-validation within each of 100 runs of the SVM, one pair of subjects (1 MCIc and 1 MCIs) was iteratively left out for testing, with the remaining subjects being used to train the model. Leave-one-out cross-validation accuracies were averaged across the 100 runs. The SVM's objective function was minimized with stochastic gradient descent. The regularization term strength of the linear classifier, lambda, was set at its default value of 1/n, where n is the training sample size. To test whether classification accuracies were significantly above chance, we performed a permutation test by shuffling the labels of the MCIc and MCIs subjects 100 times before running the classification analyses. In this case, we obtained a null distribution whose median accuracy was 0.50 and not significantly different from chance. p-values reported correspond to the proportion of values of the null distribution that exceeded the classifier's accuracy.

Next, we employed SVM-based recursive feature elimination (RFE) on balanced classes to find a minimal set of features that provided the highest generalization accuracy; details regarding the RFE technique can be found in [11]. Briefly, RFE is an iterative technique that repeatedly retrains the SVM, eliminating a subset of features with the lowest SVM weights on each iteration. The process is repeated until all but one of the features remains, following which the minimal set of features that provide the maximum generalization accuracy are identified. This procedure was repeated 75 times for random groupings of training and testing data, and generalization accuracy was averaged across these runs.

**Predicting Clinical Dementia Rating (CDR) scores based on GPFA features**. To predict the CDR-SOB scores of the subjects based on their fMRI GPFA dimensions, we performed *lasso* regression wherein each feature set – either power spectral density features (2464x1) or lagged and partial correlation features, concatenated (8855x1=(5929+2926)x1) – was the predictor variable and the CDR-SOB score was the response variables. A lasso regression model (L1 norm penalty) enabled penalizing over-fitting, as the number of features far exceeded the number of scans. We confirmed that the linear model was an appropriate fit by testing for normality of residuals. For fitting the lasso model, we pooled the data (fMRI and behavioral data) across all MCI patients (23 MCIc and 72 MCIs). One MCI subject with corrupted imaging data and two subjects for whom the CDR-SOB scores were not available were excluded from this analysis. Thus, we analyzed a total of 92 MCI patients' data. Prediction accuracy (correlations) was assessed with a k-fold cross validation approach, with values of k = 2,4,23,46 and 92. For the training data, regression coefficients were computed for 50 $\lambda$ values chosen in a geometric progression (lasso function in Matlab). Of these, we selected the regression coefficients corresponding to the $\lambda$ which generated the least mean square error for

predictions on the training data. Using these coefficients, the CDR-SOB score for the left out test fold was predicted. The correlation between the actual and predicted CDR-SOB scores was computed with Pearson correlation (*corr* function of Matlab).

## 5 Reproducing the results

Our study used two publicly available functional MRI datasets.

i) fMRI data from 1000 healthy participants from the Human Connectome Project (HCP) database is publicly available at:
`https://www.humanconnectome.org/study/hcp-young-adult/document/`
`1200-subjects-data-release`

Subjects' de-identified data were analyzed in this paper, and their HCP database id numbers are reported in SI Table S1.

ii) fMRI data from 95 MCI patients from the Alzheimers Disease Neuroimaging Initiative (ADNI) database is publicly available at: `http://adni.loni.usc.edu/data-samples/access-data/`

Subject labels (as cognitively normal, MCI or AD) are also available with documentation that accompanies the data.

Code for reproducing the results reported here is available at:
`https://figshare.com/s/22d2462b33494d79f689`

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

Figure S1: **Spatial maps and trajectories for the most representative latent components A** Spatial maps of the most representative latent dimension for the working memory task. (Top) Lateral views of the left and right hemispheres. For clarity, spatial maps have been shown separately for positive (top) and negative(bottom) values. (Middle) Time series for the corresponding latent dimensions for each task. Gray: time series for individual subjects; yellow: average time series of template subjects. (Bottom) Trajectories showing the joint activity for the most representative (x-axis) and second most representative (y-axis) latent dimensions for each task. Yellow and gray traces show average trajectories of template and test data, respectively. Red shading: normalized distribution of occupancy of the joint activity in the GPFA space spanned by these latent dimensions (n=900 subjects). Timescales corresponding to each dimension are marked along the respective axes. **B, C, D, E, F**. Same as in panel **A** but for the language (L), motor (M), gambling (G), social cognition (S), relational processing (R) and emotion processing (E) tasks, respectively.

Figure S2: **Distribution of characteristic timescales and synchronization indices across all seven tasks and resting state.** For each panel except the last (resting state), the top sub-panel depicts the distribution (histogram) of characteristic timescales. The lower sub-panel depicts the synchronization index – the average correlation among time series across subjects for each GPFA latent dimension – as a function of the characteristic timescale for that dimension. Computing the synchronization index across subjects is not meaningful for resting state scans, because resting brain fluctuations are sporadic and not synchronized across subjects. Other conventions are the same as in Fig. 2C (main text).

Figure S3: **Control analyses. A-C** Classification accuracies with an alternative parcellation. **A** Same as in Fig. 1B (main text), but determining optimal dimensions based on the Shirer et al [3] 90-node parcellation. **B** Same as in Fig. 2B (main text), but confusion matrix with GPFA trajectories extracted from the Shirer et al 90-node parcellation. **C** Comparison of task classification accuracies using GPFA trajectories and ROI time series, for Shirer et al parcellation. Color code is the same as in panel A. **D** Average accuracy of classification between sub-tasks for each task using GPFA latent time series (see SI section 2 for details). **E** Comparing latent trajectories between orthonormalized-GPFA dimensions (left column) and PCA (right column), for two representative tasks: motor (top), and social processing (bottom). Other conventions are the same as in Fig. 3A (main text).

Figure S4: **Slow oscillations in GPFA latents** Oscillatory component of GPFA spectra obtained following removal of the fractal component with IRASA [4], across all 7 tasks and resting state.

Figure S5: **Classification of MCI patient scans** GPFA-based partial correlation connectivity features that maximally distinguish (features with ranks 2 and 3) MCIc from MCIs patients. Dashed oval: fronto-insular cortex; Solid circles: anterior cingulate cortex .

Figure S6: **Spatial maps of a potential artifact dimension** Spatial maps of a latent dimension whose timeseries exhibited among the lowest values of the synchronization index across all tasks. The map is nearly identical across tasks, and resembles high frequency spatial noise, possibly due to scanner field inhomogeneities. Top and bottom rows: Lateral views of the left and right hemispheres, respectively. For clarity, spatial maps have been limited to positive and negative values in the top and bottom rows, respectively.

Table S1: Unique identifiers of 1000 subjects whose scans were analyzed from the HCP database. Red ids denote subjects in the template dataset.

| | | | | | | | | | | | | | | | | | | | |
|---|---|---|---|---|---|---|---|---|---|---|---|---|---|---|---|---|---|---|---|
| 492754 | 120212 | 100206 | 111514 | 130417 | 140925 | 152225 | 162733 | 175136 | 188145 | 200109 | 211619 | 285345 | 353740 | 429040 | 586460 | 671855 | 753251 | 837964 | 910241 |
| 495255 | 120515 | 100307 | 111716 | 130518 | 141119 | 152427 | 162935 | 175237 | 188347 | 200210 | 211720 | 285446 | 355239 | 432332 | 587664 | 672756 | 756055 | 841349 | 910443 |
| 497865 | 120717 | 100408 | 112112 | 130619 | 141422 | 152831 | 163129 | 175338 | 188448 | 200311 | 211821 | 286347 | 356948 | 433839 | 588565 | 673455 | 757764 | 843151 | 911849 |
| 499566 | 101915 | 100610 | 112314 | 130720 | 141826 | 153025 | 163331 | 175439 | 188549 | 200513 | 211922 | 286650 | 358144 | 436239 | 589567 | 675661 | 759869 | 844961 | 912447 |
| 500222 | 121416 | 101006 | 112516 | 130821 | 142828 | 153126 | 163432 | 175742 | 188751 | 200614 | 212015 | 287248 | 360030 | 436845 | 590047 | 677766 | 761957 | 845458 | 917255 |
| 114621 | 121618 | 101107 | 112920 | 130922 | 143224 | 153227 | 163836 | 176037 | 189349 | 200917 | 212116 | 290136 | 361234 | 441939 | 592455 | 677968 | 763557 | 849264 | 917558 |
| 506234 | 121921 | 102008 | 113215 | 131217 | 143325 | 153429 | 164030 | 176239 | 189450 | 201111 | 212217 | 293748 | 361941 | 445543 | 594156 | 679568 | 765056 | 849971 | 919966 |
| 510326 | 122317 | 102109 | 113316 | 131419 | 143830 | 153631 | 164131 | 176441 | 189652 | 201414 | 212318 | 295146 | 365343 | 448347 | 597869 | 679770 | 765864 | 852455 | 922854 |
| 512835 | 122620 | 102311 | 113619 | 131722 | 144125 | 153833 | 164636 | 176542 | 190031 | 201515 | 212419 | 297655 | 366042 | 449753 | 598568 | 680250 | 767464 | 856463 | 923755 |
| 513736 | 122822 | 102513 | 113922 | 131823 | 144428 | 153934 | 164939 | 176845 | 191033 | 201818 | 212823 | 298051 | 366446 | 453441 | 599065 | 680452 | 769064 | 856766 | 926862 |
| 517239 | 123117 | 102614 | 114116 | 131924 | 144731 | 154229 | 165436 | 177140 | 191235 | 202113 | 213017 | 298455 | 368753 | 453542 | 599469 | 680957 | 770352 | 856968 | 927359 |
| 519950 | 123420 | 102715 | 114217 | 132017 | 144832 | 154330 | 165638 | 177241 | 191336 | 202719 | 213421 | 299154 | 371843 | 454140 | 599671 | 683256 | 771354 | 857263 | 930449 |
| 520228 | 123521 | 102816 | 114318 | 133019 | 144933 | 154431 | 165941 | 177645 | 191841 | 202820 | 213522 | 299760 | 376247 | 456346 | 601127 | 686969 | 773257 | 859671 | 932554 |
| 522434 | 165032 | 103111 | 114419 | 133625 | 145127 | 154532 | 166438 | 178142 | 191942 | 203418 | 214019 | 300618 | 377451 | 459453 | 604537 | 687163 | 774663 | 861456 | 933253 |
| 523032 | 165840 | 103212 | 115724 | 133827 | 145632 | 154734 | 166640 | 178243 | 192035 | 203923 | 214221 | 300719 | 378756 | 461743 | 609143 | 688569 | 779370 | 865363 | 942658 |
| 114823 | 185846 | 103414 | 117021 | 133928 | 145834 | 154835 | 167036 | 178647 | 192136 | 204016 | 214423 | 303119 | 378857 | 463040 | 611938 | 690152 | 782561 | 867468 | 943862 |
| 524135 | 223929 | 103515 | 118831 | 134021 | 146129 | 154936 | 167238 | 178748 | 192237 | 204218 | 214524 | 303624 | 379657 | 465852 | 613538 | 692964 | 783462 | 869472 | 947668 |
| 525541 | 224022 | 103818 | 119025 | 134223 | 146331 | 155231 | 167440 | 178849 | 192439 | 204319 | 214726 | 304020 | 380036 | 467351 | 614439 | 693764 | 784565 | 870861 | 951457 |
| 529549 | 227432 | 104012 | 120414 | 134324 | 146432 | 155635 | 167743 | 178950 | 192540 | 204420 | 217126 | 304727 | 381038 | 468050 | 615744 | 694362 | 786569 | 871762 | 952863 |
| 529953 | 228434 | 104416 | 122418 | 134425 | 146533 | 155938 | 168139 | 179245 | 192641 | 204521 | 217429 | 305830 | 381543 | 469961 | 616645 | 695768 | 788674 | 871964 | 955465 |
| 530635 | 231928 | 104820 | 123723 | 134728 | 146735 | 156031 | 168240 | 179346 | 192843 | 204622 | 219231 | 307127 | 382242 | 473952 | 617748 | 698168 | 788876 | 872562 | 957974 |
| 531536 | 233326 | 105014 | 123824 | 134829 | 146836 | 156233 | 168341 | 180129 | 193239 | 205119 | 220721 | 308129 | 385046 | 475855 | 618952 | 700634 | 789373 | 872764 | 958976 |
| 536647 | 236130 | 105115 | 123925 | 135124 | 146937 | 156334 | 168745 | 180230 | 193845 | 205220 | 221319 | 308331 | 385450 | 479762 | 620434 | 701535 | 792564 | 873968 | 959574 |
| 540436 | 237334 | 105216 | 124220 | 135225 | 147030 | 156435 | 168947 | 180432 | 194140 | 205725 | 227533 | 309636 | 386250 | 480141 | 622236 | 702133 | 792766 | 877269 | 962058 |
| 541943 | 289555 | 105620 | 124422 | 135528 | 147636 | 156536 | 169040 | 180533 | 194443 | 205826 | 238033 | 310621 | 387959 | 481042 | 623844 | 704238 | 792867 | 878776 | 965367 |
| 545345 | 329440 | 105923 | 124624 | 135629 | 147737 | 156637 | 169444 | 180735 | 194645 | 206222 | 239136 | 311320 | 389357 | 481951 | 626648 | 705341 | 793465 | 878877 | 965771 |
| 114924 | 552544 | 106016 | 124826 | 135730 | 148032 | 157336 | 169545 | 180836 | 194746 | 206323 | 239944 | 314225 | 390645 | 485757 | 627549 | 707749 | 800941 | 880157 | 966975 |
| 548250 | 553344 | 106319 | 125222 | 135932 | 148133 | 157437 | 169747 | 180937 | 194847 | 206525 | 245333 | 316633 | 391748 | 486759 | 627852 | 709551 | 802844 | 882161 | 969476 |
| 115017 | 555348 | 106521 | 125424 | 136227 | 148335 | 157942 | 169949 | 181131 | 195041 | 206727 | 246133 | 316835 | 392447 | 510225 | 628248 | 715041 | 803240 | 884064 | 970764 |
| 115219 | 555651 | 106824 | 125525 | 136631 | 148436 | 158035 | 170631 | 181232 | 195445 | 206828 | 248339 | 317332 | 392750 | 513130 | 633847 | 715950 | 809252 | 885975 | 971160 |
| 115320 | 557857 | 107018 | 126325 | 136732 | 148840 | 158136 | 170934 | 181636 | 195649 | 206929 | 249947 | 318637 | 393247 | 516742 | 634748 | 720337 | 810843 | 886674 | 972566 |
| 115825 | 559053 | 107321 | 126426 | 136833 | 148941 | 158338 | 171330 | 182436 | 195950 | 207123 | 250427 | 320826 | 393550 | 518746 | 635245 | 723141 | 812746 | 887373 | 973770 |
| 101309 | 561242 | 107422 | 126628 | 137027 | 149236 | 158540 | 171532 | 182739 | 196144 | 207426 | 250932 | 321323 | 394956 | 519647 | 638049 | 724446 | 814548 | 888678 | 978578 |
| 116221 | 561444 | 107725 | 127226 | 137128 | 149337 | 158843 | 171633 | 183034 | 196346 | 208024 | 251833 | 322224 | 395251 | 541640 | 644044 | 725751 | 814649 | 889579 | 979984 |
| 116524 | 562345 | 108020 | 127327 | 137229 | 149539 | 159138 | 172029 | 183337 | 196750 | 208125 | 255639 | 325129 | 395756 | 550439 | 644246 | 727553 | 815247 | 891667 | 983773 |
| 116726 | 562446 | 108121 | 127630 | 137532 | 149741 | 159239 | 172130 | 183741 | 196851 | 208226 | 255740 | 329844 | 395958 | 552241 | 645450 | 728454 | 816653 | 894067 | 984472 |
| 117122 | 565452 | 108222 | 127832 | 137633 | 149842 | 159340 | 172332 | 185038 | 197348 | 208327 | 256540 | 330324 | 397154 | 555954 | 645551 | 729557 | 818455 | 894673 | 987074 |
| 117324 | 566454 | 108323 | 127933 | 137936 | 150524 | 159441 | 172433 | 185139 | 197550 | 209127 | 257542 | 333330 | 397760 | 558657 | 647858 | 731140 | 818859 | 894774 | 987983 |
| 117930 | 567052 | 108525 | 128026 | 138130 | 150625 | 159744 | 172534 | 185341 | 198047 | 209228 | 257845 | 334635 | 397861 | 558960 | 654350 | 732243 | 820745 | 896778 | 989987 |
| 118023 | 567961 | 108828 | 128127 | 138231 | 150726 | 159946 | 172938 | 185442 | 198249 | 209329 | 257946 | 336841 | 401422 | 559457 | 654552 | 734045 | 825048 | 896879 | 990366 |
| 118124 | 568963 | 109123 | 128632 | 138332 | 150928 | 160123 | 173334 | 185947 | 198350 | 209834 | 263436 | 339847 | 406432 | 561949 | 654754 | 735148 | 825553 | 898176 | 991267 |
| 118225 | 570243 | 109325 | 128935 | 138534 | 151021 | 160729 | 173435 | 186040 | 198451 | 209935 | 268749 | 341834 | 406836 | 567759 | 656253 | 737960 | 826554 | 899885 | 992673 |
| 118528 | 571144 | 109830 | 129028 | 138837 | 151223 | 160830 | 173536 | 186141 | 198653 | 210011 | 268850 | 342129 | 412528 | 578057 | 656657 | 742549 | 826353 | 901038 | 992774 |
| 101410 | 572045 | 110007 | 129129 | 139233 | 151324 | 161327 | 173637 | 186444 | 198855 | 210112 | 270332 | 346137 | 413934 | 580650 | 657659 | 744553 | 826454 | 901139 | 993675 |
| 118730 | 573249 | 110411 | 129331 | 139435 | 151425 | 161630 | 173738 | 186545 | 199150 | 210415 | 274542 | 346945 | 414229 | 580751 | 660951 | 748258 | 828862 | 901442 | 994273 |
| 118932 | 573451 | 110613 | 129634 | 139637 | 151526 | 161731 | 173839 | 186848 | 199352 | 210617 | 275645 | 348545 | 415837 | 581349 | 662551 | 749058 | 832651 | 902242 | 996782 |
| 119126 | 576255 | 111009 | 129937 | 139839 | 151627 | 161832 | 173940 | 187143 | 199453 | 211114 | 280739 | 349244 | 419239 | 581450 | 663755 | 749361 | 833148 | 904044 | 668361 |
| 119732 | 579665 | 111211 | 130013 | 140117 | 151728 | 162026 | 174437 | 187345 | 199655 | 211215 | 281135 | 350330 | 421226 | 583858 | 664757 | 751348 | 833249 | 905147 | 685058 |
| 119833 | 579867 | 111312 | 130114 | 140319 | 151829 | 162228 | 174841 | 187547 | 199958 | 211316 | 283543 | 352132 | 422632 | 585256 | 665254 | 751550 | 835657 | 907656 | 872158 |
| 120111 | 580044 | 111413 | 130316 | 140824 | 151930 | 162329 | 175035 | 187850 | 200008 | 211417 | 284646 | 352738 | 424939 | 585862 | 667056 | 753150 | 837560 | 908860 | 937160 |

Table S2: Description of tasks performed inside the fMRI scanner.

| Task Name/Key | Description | Subtasks |
|---|---|---|
| Resting state (rs) | Resting state with eyes open, relaxed fixation. | _ |
| Working Memory (W) | N-back working memory, body parts, tools, places | 0bk: 0-back working memory blocks<br>2bk: 2-back working memory blocks |
| Language (L) | Sentences, stories, mental arithmetic(auditory) | Math: blocks for solving math problems<br>Story: blocks with story-based questions |
| Motor (M) | Hand, foot, tongue movements | Hand: left, right finger movement blocks<br>Foot: left, right toe movement blocks |
| Social Cognition (S) | Interpret social vs. random interaction | Mental: blocks with interacting shapes<br>Random: blocks with random movement |
| Gambling (G) | Reward, punishment, decision making | Win: blocks with mostly reward<br>Loss: blocks with mostly loss |
| Relational Processing (R) | Higher-order cognition | Relation: relational processing blocks<br>Match: shape, texture matching blocks |
| Emotion Processing (E) | Valence judgements (faces) and shape recognition | Fear: emotional face matching blocks<br>Neutral: shape matching blocks |

Table S3: Behavioral scores and descriptions.

| Index | Abbreviation | Age: adj/unadj | Description |
|---|---|---|---|
| 1 | MMSE_Score | - | Mini Mental Status Exam Total Score |
| 2 | PSQI_Score | - | Pittsburgh Sleep Questionnaire Total Score |
| 3 | ReadEng | Adjusted | NIH Toolbox Oral Reading Recognition Test Score |
| 4 | PicVocab | Adjusted | NIH Toolbox Picture Vocabulary Test Score |
| 5 | Endurance | Adjusted | NIH Toolbox 2-minute Walk Endurance Test Score |
| 6 | Strength | Adjusted | NIH Toolbox Grip Strength Test Score |
| 7 | GaitSpeed_Comp | - | NIH Toolbox 4-Meter Walk Gait Speed Test: Computed Score |
| 8 | Dexterity | Adjusted | NIH Toolbox 9-hole Pegboard Dexterity Test Score |
| 9 | PicSeq | Adjusted | NIH Toolbox Picture Sequence Memory Test Score |
| 10 | CardSort | Adjusted | NIH Toolbox Dimensional Change Card Sort Test Score |
| 11 | Flanker | Adjusted | NIH Toolbox Flanker Inhibitory Control and Attention Test Score |
| 12 | ProcSpeed | Adjusted | NIH Toolbox Pattern Comparison Processing Speed Test Score |
| 13 | DDisc_200 | - | Delay Discounting: Area Under the Curve for Discounting of $200 |
| 14 | DDisc_40K | - | Delay Discounting: Area Under the Curve for Discounting of $40,000 |
| 15 | FluInt_CR | - | Penn Progressive Matrices: Number of Correct Responses |
| 16 | FluInt_SI | - | Penn Progressive Matrices: Total Skipped Items |
| 17 | FluInt _RTCR | - | Penn Progressive Matrices: Median Reaction Time for Correct Responses |
| 18 | VSPLOT_TC | - | Variable Short Penn Line Orientation: Total Number Correct |
| 19 | VSPLOT_OFF | - | Variable Short Penn Line Orientation: Total Positions Off for All Trials |
| 20 | VSPLOT_CRTE | - | Variable Short Penn Line Orientation: Median Reaction Time divided by Expected Number of Clicks for Correct Trials |
| 21 | SCPT_SEN | - | Short Penn Continuous Performance Test: Sensitivity |
| 22 | SCPT_SPEC | - | Short Penn Continuous Performance Test: Specificity |
| 23 | SCPT_TPRT | - | Short Penn CPT Median Response Time for True Positive Responses |
| 24 | SCPT_LRNR | - | Short Penn Continuous Performance Test: Longest Run of Non-Responses |
| 25 | ListSort | Adjusted | NIH Toolbox List Sorting Working Memory Test Score |
| 26 | IWRD_TOT | - | Penn Word Memory Test: Total Number of Correct Responses |
| 27 | IWRD_RTC | - | Penn Word Memory Test: Median Reaction Time for Correct Responses |