[Reviews · NeurIPS 2019]

Reviewer 1



Originality: The tasks or methods are not new exactly, as they have been applied in electrophysiology (reference 8 in the manuscript), but they are new as applied to fMRI. This is therefore important. Quality: Experiments and application of methods are entirely sound, gpfa method is compared to other PCA methods and is applied to large datasets. Authors acknowledge prospective work needs to be done to draw conclusions in their AD results. Clarity: The submission is well written and organized. Significance: The significance is perhaps the weakest area judged because authors take a known method from a different modality (GPFA has been used extensively in electrophysiology which the authors acknowledge) and apply it to fMRI. To my knowledge the authors have been able to reach conclusions not previously reachable with other dimensionality reduction methods in fMRI datasets. It is important to apply methods from other complementary fields to reach broader conclusions about brain dynamics even if not inventing an entirely new algorithm to do so.

Reviewer 2



The authors provide a new integrated analysis approach (allowing for simultaneous dimensionality reduction and the possibility of de-noising/artifact correction) to assess slow and infra-slow fluctuations of functional MRI data. They evaluate their approach in a very representative sample and show its potential utility by decoding the task that participants were asked to perform, while being scanned, as well as by predicting behavioral scores from the newly derived latent components as well as clinically-relevant outcomes in a clinical sample. In the following sections, I provide specific feedback with respect to originality, quality, clarity and significance. I hope you will find my comments helpful and constructive. Originality To my knowledge the proposed approach is a novel and innovative way of assessing (task-related or task-free) functional connectivity in the brain in a data-driven manner. Due to its inherent flexibility (e.g. the possibility of incorporating different sources of noise and other covariates of interest, as well as explicitly specifying different kernels), I believe it constitutes a very promising approach that can be extended to address a wide range of questions in the future. While reading the introduction I had the impression that the authors slightly miss-represent the current level of understanding with respect to slow fluctuations or the BOLD response in the brain. While it is true that there are open questions, also with respect to neurovascular coupling, there are studies that have shed light on the nature of the BOLD response (most notably: Logothetis et al., 2001, Neurophysiological investigation of the basis of the fMRI signal, Nature; or Lee et al., 2010, Global and local fMRI signals driven by neurons defined optogenetically by type and wiring, Nature). Claims like ‘For the aforementioned reasons, however, it is unclear whether slow and infra-slow brain dynamics, as measured with fMRI, are relevant for cognitive function.’ (L. 30) seem to be a bit of an overstatement. Especially since I believe that BOLD responses have been related to cognitive function in thousands of studies and these bold responses also fall into the category of what the authors term slow fluctuations (since a canonical HRF has a duration of ~25s-30s corresponds to 0.04-0.033 Hz). I fail to see how these dynamics are qualitatively different from the dynamics the authors study. I also fail to see how one would be able to interpret any dynamics faster than 1Hz given that a typical sampling rate is about 2s (0.5Hz) and the corresponding Nyquist frequency is 0.25 Hz. Can one expect any meaningful sampling of higher frequencies at all in an fMRI study? Despite these issues, I believe the study to make a very original and useful contribution, but I would recommend framing it rather as a new method to study functional connectivity in the brain, with less emphasis on ‘slow dynamics’, which suggests to readers that this is something fundamentally different than what has been done before. Clarity & Quality The paper is written clearly and concisely. I especially appreciate the effort the authors made to show robustness with respect to a different parcellation, comparisons with known strategies, as well as the prediction applications. In the following, I have a few open questions and suggestions for further improvement: One aspect that could be improved is the section about predicting MCIc, it requires a more detailed description of the procedure (at least in the supplementary material). It is unclear to me how the authors accounted for the class-imbalance. Has there been class-weighing, over- or under-sampling? As I understood the supplementary material it seems to me that the MCIs group was over-sampled, however, the MCIc group constitutes the less frequent class. Would you care to elaborate on that? It is also unclear to me why LSO-CV was chosen, as it is prone to overfit. Would a different CV strategy (5-fold for example) not be more appropriate? It would also be helpful to assess ‘potential artifact’ maps and show them to the reader. Do they present known artefactual spatial pattern (close to the boundaries of the brain or ventricles)? Also is the assumption ‘For example, artifacts, such as scanner noise or physiological noise, are unlikely to be synchronized across subjects, whereas dimensions strongly locked to salient aspects of the task would be highly synchronized.’ (L. 154 ff.) indeed correct, since for example behavioral responses that cause motion artifacts may be synchronized across subjects and occur on a slow time scale (e.g. one motion in each trial of the task)? In Figure 2D and 2E it seems as if PCA series-based task classification worked as well as GPFA-series-based classification, secondly, the same seems to be true for GPFA-spectra vs. ROI-spectra. Would it be more accurate to state the GPFA method was at-least as good as conventional methods (rather than stating that it outperformed the conventional methods (L184)? How did you assess significance here? L.186 paragraph on predicting behavioral measures: It would be good to inform the reader on the selection criteria of the behavioral measures you chose to predict. Were they randomly selected or based on previous literature? Also you write ‘that a majority of scores were consistently well predicted’ (L195). Arguably, a correlation of ~.2 is not a very good prediction, it may be a significant prediction, but it certainly does not explain a lot of variance in the target. Maybe this could be rephrased a bit more cautiously. Significance As I already alluded to, I believe that this work constitutes a significant methodological contribution as this approach is a novel and innovative way of assessing (task-related or task-free) functional connectivity in the brain in a data-driven manner. Additionally, the authors undergo a lot of effort to make comparisons to existing methods and validate the predictive utility. Applying this method to a clinical sample constitutes a very valuable empirical contribution in and of itself. COMMENTS TO THE AUTHORS' RESPONSE: First of all, I wanted to thank the authors for a very carefully drafted response. I appreciated that you addressed my concerns in detail. I feel that you adequately address most of my concerns. I would only like to suggest that you include your rationale for selecting behavioral measures in the supplement of the revised manuscript and I would like to clarify one comment I made regarding the implications of your work with respect to how the brain works. I agree that your methods may allow some insights into connectivity pattern that drive task-specific behavior or preceed disease. I just wanted to point out that a data-driven method of the sort that you use comes at a cost of reduced interpretability. If one explicitly models the hemodynamic response function one can make inferences on local vasculature more readily and it is somewhat simpler to relate the BOLD response to neuronal activity. Of course, assumptions that are made to model this can be wrong and this is were I see and aknowledge the added value of your approach. I also want to reiterate that for a potential clinical application (as you have shown) interpretability plays a secondary role. I just felt that this potential short coming should be aknowledged (you also run into this, when it comes to interpreting the high-frequency latents) and would like to encourage you to focus on applications such as the MCI prediction in the future, which I believe is much more valuable than classifying task (however, I understand your motivation for using this as a first toy example). I thank you very much for your great work and hope it will receive the credit it deserves.

Reviewer 3



One of the primary challenges in fMRI datasets is the slow hemodynamic response. This paper proposes to employ a Gaussian Process Factor Analysis (GPFA) approach for dealing with the mentioned problem. In practice, GPFA is applied to slowly sampled fMRI features provided by the Human Connectome Project for mapping the neural activities to a lower-dimensional space and then extracting smooth latent dynamics. This topic is interesting, and the experiments show some improved accuracy. However, the machine learning novelties and the contributions of this paper is ambiguous. The primary concerns and the minor comments are listed in the following: 1. Although extracting a new lower-dimensional feature space from neural activities (by using GPFA) can increase the accuracy of analysis, there is no causal effect to demonstrate that this improvement is related to fixing the slow hemodynamic issue! 2. Another concern here is the number of data resources. Although the number of samples is acceptable, these data are provided from the same resource. How will the performance of the proposed method be changed if we have batch effects such that fMRI datasets are collected from different machines or locations? 3. The proposed method must be clearly formulated. While this paper in the current format provided a long story (in both the original article and the supplementary material), it is hard to follow the main idea. 4. The limitations and applications of the proposed method must be discussed in details. 5. While noisy features are mentioned repetition through the paper, there is no empirical study to demonstrate what the effect of noise on the performance of the proposed method is? 6. Table S1 in the current format is not informative-rich. === Update after reading rebuttle === Currently, the author(s) address the main concerns in the rebuttal letters, including comment 1 and 2. That is the reason that I have updated the score. However, they must present the revised version of Sections 1 and 2 (answer to comment 3). The presented answers to comments 4 to 6 also must be added to the final version of the paper.

[Author Response · NeurIPS 2019]

**Author response for submission # 3770**. We thank the reviewers for their valuable feedback.

**§R1.** *limitations to using Gaussian-based methods . . .* BOLD time series were not binned into counts, but rather directly analyzed as a continuous signal. GPFA was applied without conventional pre-processing steps for spikes (binning, square-root transform, kernel smoothing). We also verified that a majority of the fMRI time series data satisfied Gaussianity assumptions. For the HCP dataset, 78% of regional time series passed the Lilliefors test for normality. For the MCI dataset, 81% passed the test for normality. These are now reported in the revision.

**§R1.** *Given the variances of BOLD counts is known to increase with the mean . . . Does a Poisson . . . correction need to be considered?* To avoid the confound of variance scaling (or changing) with the mean, observed BOLD time series were variance normalized by z-scoring (mean subtracted and divided by standard deviation) before GPFA. Future work can incorporate a Poisson correction, so that the method can be directly applied to the observed BOLD data.

**§R2.** *the authors were able to decode the tasks so well - it does not allow for . . . insights into . . . how the brain works.* Examining matrix C in the GPFA model (SI section 1, line 6), provides some insights into brain networks that contribute to dynamics at specific timescales (e.g. map in Fig. 1D, top). Classification analysis with MCI patients, provides insights into which networks, and at what timescales, may be implicated in cognitive decline (e.g. Fig 4C-D).

**§R2.** *the authors slightly miss-represent the current level of understanding . . .* We have revised the Introduction based on the reviewer's suggestions to provide an up-to-date picture.

**§R2.** *how these dynamics are qualitatively different from the dynamics the authors study.* Dynamics in our study are unlikely to be qualitatively different from those previously studied. Our approach permits quantifying the specific timescale associated with each latent dimension, which enables linking dynamics at specific timescales with behavior.

**§R2.** *how one would be able to interpret any dynamics faster than . . . Nyquist frequency [of] 0.25 Hz.* The faster timescales arise because HCP data is sampled at 0.72s (1.4 Hz). Interpreting timescales faster than 0.7 Hz is indeed challenging. GPFA latents at these timescales likely reflect fits to the residual noise after accounting for slow latents.

**§R2.** *recommend framing it rather as a new method to study functional connectivity.* We have now emphasized the novelty of the method for studying fMRI functional connectivity, and reduced the emphasis on slow dynamics.

**§R2.** *predicting MCIc, . . . requires a more detailed description . . .* We have now elaborated the description.

**§R2.** *unclear to me how the authors accounted for the class-imbalance . . . MCIs group was over-sampled?* The MCIc group (the minority class) was over-sampled. To address class imbalance, all 22 MCIc subjects, and 22 MCIs subjects sampled from the pool of 72, were used in each of 100 runs of the SVM. Each MCIs subject was sampled at least once every 4 runs. For LOO cross-validation one pair of (1 MCIc and 1 MCIs) subjects was left out for testing, with the remaining subjects being used to train the model. CV accuracies were averaged across the 100 runs.

**§R2.** *(CDR-SOB prediction). . . Would a different CV strategy (5-fold for example) not be more appropriate?* We now perform k-fold CV analysis with k=2,4,23,92 (92 subjects with CDR scores). Predicted vs observed score correlation increased steadily with more folds, perhaps because of the small sample size and bias in the training data with few folds.

**§R2.** *. . . helpful to assess 'potential artifact' maps.* We have added an SI subsection that discusses these maps in detail.

**§R2.** *motion artifacts may be synchronized across subjects. . . ?* We acknowledge this in the revised Discussion.

**§R2.** *Would it be more accurate to state the GPFA method was at-least as good as conventional methods?* We have revised the claim. Yet, only with GPFA did both spectra *and* time series provide high classification accuracies. Significance was assessed with Wilcoxon signed rank test (SI section 2, lines 94-113).

**§R2.** *inform . . . the selection criteria of the behavioral measures* Behavioral scores were selected from Alertness, Cognition, Motor categories (HCP Data Dictionary). Only scores that carried independent information were selected (e.g. sensitivity and specificity, but not true/false positive rates). Scores were selected "blind" to the prediction analyses.

**§R2.** *correlation of .2 is not a very good prediction.* We have rephrased this claim more carefully.

**§R2.** *Improvements.* Addressed above (frequencies studied in fMRI; literature overview; details about MCI prediction).

**§R3.** *1. . . . no causal effect to demonstrate that this improvement is related to fixing the slow hemodynamic issue!* We do not propose a solution that overcomes slow hemodynamics. Rather, we estimate infra-slow dynamics by applying GPFA to fMRI data (BOLD timeseries) directly, and demonstrate its relevance for cognitive state prediction.

**§R3.** *2. . . . How will the performance . . . be changed if . . . fMRI datasets are collected from different machines or locations?* Our results already incorporate this diversity. HCP data was acquired at Wash. Univ. in St. Louis. ADNI data was collected at multiple centers in the US. Scanners were of different makes including, Siemens, Philips and GE.

**§R3.** *3. The proposed method must be clearly formulated.* Main paper (section 2) and SI (section 1) are revised.

**§R3.** *4. The limitations and applications of the proposed method . . .* Limitations include assumptions on Gaussianity, linking hemodynamics to neural dynamics, and scalability for voxel-level estimation. Applications include predicting behavioral scores, classifying pathologies and identifying signatures of cognitive decline. Discussed in the revision.

**§R3.** *5. . . . what the effect of noise on the performance of the proposed method is?* GPFA reduces dimensionality by identifying latent dimensions with maximal covariance among ROIs. With an explicit noise model, GPFA estimates noise that is independent across ROIs (SI, lines 6-7). Thus, adding independent noise across ROIs generally does not affect parameter estimates. Adding correlated noise enables GPFA to factor these into a separate latent dimension.

**§R3.** *6. Table S1 . . . is not informative-rich.* Publicly available data regarding subjects (gender and age-group) have now been added in the revised SI (e.g. of 1000 subjects, 529 females, age range: 22-35 years, 10 subjects of age >36 years).

[Meta-Review · NeurIPS 2019]

This paper proposes the use of Gaussian process factor analysis (GPFA) to analyze the latent dynamics of spontaneously generated electrical activity of the brain as measured using fMRI. The paper shows that GPFA can be used to extract slowly varying latent factors that are informative for understanding cognitive processes in healthy brains and cognitive decline in diseased brains. While GPFA is not a new method and has been applied in many other areas of data analysis, it has not previously been applied to the problem of inferring latent dynamics from fMRI data. Further, as the authors show, this approach appears to yield superior performance to a range of alternative approaches on two different tasks using large, real data sets. The reviewers judged the empirical results to be highly significant as they point to new data-driven ways of understanding the relationship between brain and behavior. All reviewers were satisfied with the author response following the discussion of the paper and the consensus of the reviewers is that the paper should be accepted. The authors should be sure to incorporate the responses to the reviewers original concerns into the updated manuscript and to address any issues that remain in the final reviews.